# Major Adverse Cardiovascular Events in Coronary Type 2 Diabetic Patients: Identification of Associated Factors Using Electronic Health Records and Natural Language Processing

**DOI:** 10.3390/jcm11206004

**Published:** 2022-10-11

**Authors:** Carlos González-Juanatey, Manuel Anguita-Sánchez, Vivencio Barrios, Iván Núñez-Gil, Juan José Gómez-Doblas, Xavier García-Moll, Carlos Lafuente-Gormaz, María Jesús Rollán-Gómez, Vicente Peral-Disdier, Luis Martínez-Dolz, Miguel Rodríguez-Santamarta, Xavier Viñolas-Prat, Toni Soriano-Colomé, Roberto Muñoz-Aguilera, Ignacio Plaza, Alejandro Curcio-Ruigómez, Ernesto Orts-Soler, Javier Segovia, Víctor Fanjul, Ángel Cequier

**Affiliations:** 1Hospital Universitario Lucus Augusti, 27003 Lugo, Spain; 2Hospital Universitario Reina Sofía, Instituto Maimonides de Investigación Biomédica de Córdoba (IMIBIC), Universidad de Córdoba, 14014 Córdoba, Spain; 3Hospital Universitario Ramón y Cajal, 28034 Madrid, Spain; 4Hospital Clínico Universitario San Carlos, 28040 Madrid, Spain; 5Hospital Universitario Virgen de la Victoria CIBERCV (Centro de Investigación Biomédica en Red Enfermedades Cardiovasculares) and IBIMA (Instituto de Investigación Biomédica de Málaga), 29010 Málaga, Spain; 6Hospital Universitario Santa Creu i Sant Pau, 08041 Barcelona, Spain; 7Hospital Universitario de Albacete, 02006 Albacete, Spain; 8Hospital Universitario Río Hortega, 47012 Valladolid, Spain; 9Hospital Universitario Son Espases, 07120 Palma de Mallorca, Spain; 10Hospital Universitario y Politécnico La Fe, CIBERCV, IIS La Fe, 46026 València, Spain; 11Hospital Universitario de León, 24071 León, Spain; 12Hospital Vall d’Hebron, CIBERCV, 08035 Barcelona, Spain; 13Hospital Infanta Leonor, 28031 Madrid, Spain; 14Hospital Infanta Sofía, 28703 Madrid, Spain; 15Hospital Universitario de Fuenlabrada, 28942 Madrid, Spain; 16Hospital General Universitario de Castellón, 12004 Castellón de la Plana, Spain; 17Hospital Universitario Puerta de Hierro, 28222 Madrid, Spain; 18Savana Research, 28013 Madrid, Spain; 19Hospital Universitario de Bellvitge, IDIBELL (Instituto de Investigación Biomédica de Bellvitge), Universidad de Barcelona, 08007 Barcelona, Spain

**Keywords:** diabetes mellitus, coronary artery disease, MACE, risk factors, electronic health records, natural language processing

## Abstract

Patients with Type 2 diabetes mellitus (T2DM) and coronary artery disease (CAD) are at high risk of developing major adverse cardiovascular events (MACE). This is a multicenter, retrospective, and observational study performed in Spain aimed to characterize these patients in a real-world setting. Unstructured data from the Electronic Health Records were extracted by EHRead^®^, a technology based on Natural Language Processing and machine learning. The association between new MACE and the variables of interest were investigated by univariable and multivariable analyses. From a source population of 2,184,662 patients, we identified 4072 adults diagnosed with T2DM and CAD (62.2% male, mean age 70 ± 11). The main comorbidities observed included arterial hypertension, hyperlipidemia, and obesity, with metformin and statins being the treatments most frequently prescribed. MACE development was associated with multivessel (Hazard Ratio (HR) = 2.49) and single coronary vessel disease (HR = 1.71), transient ischemic attack (HR = 2.01), heart failure (HR = 1.32), insulin treatment (HR = 1.40), and percutaneous coronary intervention (PCI) (HR = 2.27), whilst statins (HR = 0.73) were associated with a lower risk of MACE occurrence. In conclusion, we found six risk factors associated with the development of MACE which were related with cardiovascular diseases and T2DM severity, and treatment with statins was identified as a protective factor for new MACE in this study.

## 1. Introduction

Diabetes mellitus (DM) is a key public health problem worldwide associated with the development of complications such as cardiovascular comorbidities, which include coronary artery disease (CAD) [1,2,3]. It was listed as the underlying cause of mortality in >85,000 deaths in the United States in 2019 [2], consisting mainly of cardiovascular complications and CAD development, the main causes of death and disability in patients with type 2 diabetes mellitus (T2DM) [4]. In addition, DM has been associated with a 2 to 4-fold increased risk of death due to heart disease [5].

Given the clinical burden that cardiovascular complications have on patients with T2DM, there has been an increased focus on the joint management of both entities, T2DM and cardiovascular disease (CVD). Furthermore, during the last decade, there has been increasing pressure from regulatory agencies for antidiabetic treatments to demonstrate cardiovascular safety and benefits, especially for major adverse cardiovascular events (MACE) [6]. However, patients included in controlled trials are generally selected on the basis of protocol eligibility and, therefore, do not fully represent the entire population encountered in clinical practice [7]. In this regard, real-world-based studies are needed to improve diagnostic and therapeutic approaches to palliate the burden of disease by reducing cardiovascular events.

The present study aimed to use population-based data to provide real-world insights into the clinical characterization and treatment management of diabetic patients who develop CAD in Spain. Moreover, this study investigated the probability of developing MACE after presentation of CAD in T2DM patients and the potential risk or protective factors associated with its occurrence within this population. To do so, we used EHRead^®^ (Madrid, Spain), a technology that applies Natural Language Processing (NLP) and machine learning, to extract, organize, and analyze the unstructured clinical information that health professionals register in patients’ electronic health records (EHRs) [8,9,10,11,12,13].

## 2. Materials and Methods

### 2.1. Study Design

This was a real-world, multicenter, retrospective, and observational study (ACORDE: “Assessment of medical management in CORonary DiabEtic Type 2 patients at high risk of cardiovascular events”) based on the secondary use of the unstructured data captured in the EHRs. A cross-sectional analysis of all patients was performed at the index date, defined as the first time in the study period when the patient fulfilled all inclusion and no exclusion criteria. At this point, demographics, comorbidities, vital signs, general characteristics of T2DM and CAD and treatments were evaluated. The cumulative incidence of MACE, defined as previously described by the presence of myocardial infarction, stroke, hospitalization for unstable angina, and urgent coronary revascularization [14], was analyzed during the follow-up. Although included in this definition, all-cause or cardiovascular death was not analyzed as, due to the nature of the data source, their occurrence was not accurately identifiable. The follow-up period was defined as the period of time spanned from index date to the last EHR available for each patient within the study period.

### 2.2. Data Source

The data source was the free-text information within the EHRs of 12 representative hospitals from six major regions in Spain: Madrid (Hospital Universitario de Fuenlabrada, Hospital Universitario Infanta Sofía, Hospital Universitario Infanta Leonor, Hospital Universitario Puerta de Hierro), Catalonia (Hospial Universitari Vall d’Hebron, Hospital de la Santa Creu i Sant Pau), Valencia (Hospital Universitari i Politècnic La Fe, Hospital General Universitario de Castellón), Balearic Islands (Hospital Universitari Son Espases), Castilla La Mancha (Complejo Hospitalario Universitario de Albacete), and Castilla y León (Hospital Universitario Río Hortega, Hospital Universitario de León). Data entry in outpatient clinical reports, discharge reports, emergency reports, prescriptions, and other medical reports, were collected from all available services and departments in each participating site, including inpatient hospital, outpatient hospital, and emergency room, between 1 January 2014 and 31 December 2018.

### 2.3. Study Subjects

The source population of the study comprised all adult patients with available EHRs in the participating hospitals within the study period with at least 6 months of follow-up. Patients with T2DM and a diagnosis of CAD were included. These two entities were considered when they were contained in the unstructured, free-text information in the EHRs based on clinical diagnosis. T2DM were also considered when documented ongoing use of glucose-lowering drugs (oral hypoglycemic agents) was found for at least 6 months. CAD was considered if there was evidence of stenosis ≥ 50% of at least 1 coronary artery but without a history of having a previous myocardial infarction or stroke, and without planned coronary, cerebrovascular, or peripheral arterial revascularization. Patients with prior myocardial infarction or stroke, history of liver cirrhosis or liver cancer, intracranial bleeding, renal failure requiring dialysis, or ongoing treatment with anticoagulant medication at index date or non-available follow-up information spanning at least 6 months were excluded from the study.

### 2.4. Extraction of the Unstructured Free-Text from EHRs

Unstructured clinical data were extracted using the EHRead^®^ technology, which uses NLP, machine learning, and deep learning techniques for extracting free text from de-identified and processed EHR and translates it into a study database. EHRead^®^ technology have been described elsewhere [9,10,12,13,15,16,17,18,19].

Using the information obtained from this process, a statistical model was generated to describe the demographic and clinical characteristics of the population with T2DM and CAD. Information was analyzed and expressed by means of concepts that contain the most significant information in the text. The terminology considered by EHRead^®^ includes codes, concepts, synonyms, and definitions used in clinical documentation and is based on SNOMED CT [20,21].

### 2.5. External Validation of EHRead^®^ Performance

EHRead^®^ performance was assessed to evaluate the ability of the technology to identify mentions of coronary disease and its related clinical variables within patients’ EHRs [9,12,13,22]. First, a comparison between a physician-annotated set of EHRs (i.e., “gold standard”) and the “output” of EHRead^®^ upon reading that same set of EHRs was carried out independently at each hospital site [9]. Thus, two designated expert physicians from each hospital annotated a set of randomly selected records while a third physician reviewed the annotations made and resolved any possible discrepancies among the two site physicians. The “output vs. gold standard” comparison was reflected in the standard evaluation metrics, including precision, recall, and their harmonic mean (F1-score). As previously described, we demonstrated a high performance of EHRead^®^ at identifying records that contain mentions of “Coronary disease” in the target population with a precision, recall, and F1-score of 0.89, 0.73, and 0.80, respectively. Moreover, the evaluation yielded a F1-score of above 90% in most of the variables analyzed [9,10].

### 2.6. Statistical Data Analyses

Categorical variables were described via frequency tables; whereas numerical variables were presented using summary tables that included the mean, standard deviation (SD), median, and quartiles (Q1, Q3). Missing data were handled according to the nature of the data collection process, thus lack of information (i.e., unavailable data in patients’ EHRs) was considered a “true zero” for binary variables (e.g., absence of a comorbidity) but was treated as missing data for numerical variables (e.g., laboratory values). No missing data imputation was performed. Cumulative incidence of new cases of MACE was estimated through the Kaplan–Meier approach. Both univariable and multivariable analyses were performed to investigate the association between the occurrence of new MACE and the variables of interest. Variables with >20% missing values or with zero variance were excluded. In the univariable analysis, a Cox proportional hazard (PH) regression model was fitted to the study population for each variable at baseline. In the multivariable approach, a data-driven variable and model selection was performed. First, a Cox PH model was fitted to the study population using all eligible study variables. Then, we carried out backward stepwise selection (i.e., the least informative variables were excluded one at a time, and new models were fitted in a recursive manner). We used the Bayesian information criterion (BIC) to reach a model with the optimal explanatory variables. Significant differences were considered when *p* < 0.05. All analyses were performed using “R” software, version 4.0.2 (The R Foundation for Statistical Computing, Vienna, Austria).

### 2.7. Ethical Considerations and Study Approval

This study was classified as a “non-post-authorization study” by the Spanish Agency of Medicines and Health Products (AEMPS) and was approved by the Institutional Review Board (IRB) of each participating hospital. All methods and analyses were conducted in compliance with local legal and regulatory requirements, as well as generally accepted research practices described in the Helsinki Declaration in its latest edition, and Good Pharmacoepidemiology Practices. Data were analyzed from de-identified EHRs, which were aggregated in an irreversibly dissociated manner. Thus, individual patient consent was not required in the study.

## 3. Results

### 3.1. Study Population

The source population of the study comprised a total of 2,184,662 adult patients with available EHRs in the hospitals of interest including 217,632 (10%) patients with a confirmed diagnosis of T2DM. The study population only included adult patients with T2DM and stable CAD diagnosis with no previous history of myocardial infarction and stroke, which yielded to a final population of 4072 patients. In total, 37.4% of the patients presented insulin-dependent diabetes. Type of CAD was identified in 60% of the patients. Of which, 833 of the patients (34.21% of the patients with CAD type identified within the EHR) had single coronary vessel disease and 1,602 (65.79%) multivessel coronary disease. Patients were followed up for a median of 33.6 months (IQ1-3: 19.5–47.1).

### 3.2. Demographic and Clinical Characteristics

The sociodemographic and comorbidities of the study population at index date are summarized in Table 1 and Figure 1. As shown, the mean age of the patients was 70 ± 11 years, with two thirds being male (62.2%) and 17.5% current smokers (Table 1).

The most common cardiovascular comorbidities reported in the target population included arterial hypertension (84.65%), and heart valve disease (38.51%) (Figure 1A). In regard to other comorbidities observed, the most frequent disorder was hyperlipidemia (40.91%), followed by obesity (32.60%) (Figure 1B).

As shown in Appendix A, our study population had clinical parameters consistent with metabolic abnormalities including elevated glucose, body mass index, lipids parameters and hemoglobin A1c values.

### 3.3. Disease Management

Treatment strategies in our study population at index date included pharmacological management and interventional approaches.

Data regarding pharmacological treatment for the management of patients diagnosed with T2DM and CAD are shown in Figure 2.

Metformin was identified as the most frequently prescribed antidiabetic drug (77.6%), followed by insulin/analogs (25%), sulfonylureas (21.64%), dipeptidyl peptidase-4 inhibitors (iDPP4) (20.83%), and glinides (12.45%). Other treatments such as statins (79.15%) and antiplatelet therapy, mainly ASA (64.0%), were commonly prescribed according to the available information within the EHRs.

Regarding interventional procedures, we observed that a total of 1574 patients (38.8%) were revascularized with percutaneous coronary interventions (PCI) and 585 patients (14.4%) were treated with coronary artery bypass graft (CABG).

### 3.4. Major Adverse Cardiovascular Events (MACE)

We evaluated the cumulative incidence of MACE, during the study follow-up.

As shown in Table 2, the probability of developing at least one MACE after 4-years follow-up was 27.09%, with it being more probable to undergo myocardial infarction (13.89%) than developing ischemic stroke (6.94%), unstable angina (4.54%), or urgent coronary revascularization (8.69%).

Cumulative incidence curves of the first MACE occurrence after study entry and detailed probability of individual components of interest over study follow-up are shown in Figure 3 and Appendix A, respectively.

The univariable analysis using Cox PH models showed that age, history of smoking, and the time of first mention of T2DM and CAD disease in the EHRs were significantly associated with the occurrence of new MACE during the follow-up period, while female gender was inversely associated with the development of MACE (*p* < 0.001) (Appendix A).

Regarding comorbidities, in general, the presence of all cardiovascular disorders was associated with the occurrence of the first MACE detected during the follow-up. Likewise, diabetic retinopathy, hypothyroidism, and COPD/asthma were found to be associated with the appearance of new MACE in the study (Appendix A). In addition, new MACE were associated with prescription of insulin (*p* < 0.001), some hypoglycemic agents including sulfonylureas (*p* = 0.008) and alpha-glucosidase (*p* = 0.049), anticoagulant and antiplatelet therapy (*p* = 0.021 and *p* < 0.001, respectively), beta blockers, calcium channel, nitrates, ranolazines, lipid lowering drugs (*p* < 0.001 each), diuretics (*p* = 0.012), and statins (*p* = 0.003) (Appendix A). Revascularization treatment, including PCI and CABG were also found to be related the occurrence of new MACE during the study period (both *p* < 0.001).

In a multivariable approach using backward stepwise selection based on BIC, we found 7 independent variables to be associated with MACE during follow-up, including coronary involvement with multivessel and single coronary vessel disease (HR = 2.49; 95% CI = 2.01–3.10; *p* < 0.001), and (HR 1.71; 95% CI = 1.34–2.18; *p* < 0.001), transient ischemic attack (HR = 2.01; 95% CI = 1.46–2.77; *p* < 0.001), and heart failure (HR = 1.32; CI 95% = 1.13–1.53; *p* < 0.001). Pharmacological treatment with insulin and interventional therapies as PCI were also found to be independently associated with the occurrence of new MACE (HR = 1.40; CI 95% = 1.21–162; *p* < 0.001 and HR = 2.27; CI 95% = 1.92–2.67; *p* < 0.001, respectively). On the contrary, treatment with statins was found to be inversely associated with MACE during follow-up (Figure 4).

## 4. Discussion

The present study uses NLP and machine learning to characterize patients with T2DM and CAD in a real-world setting in Spain. Our results showed that: (1) T2DM patients who develop CAD are predominantly male with arterial hypertension, hyperlipidemia and obesity as the most prevalent comorbidities; (2) these patients present an increasing probability of developing MACE during the study follow-up; and (3) in this population the risk factors associated with the development of MACE are related with cardiovascular diseases (multivessel or single vessel disease, PCI, transient ischemic attack, heart failure) and T2DM severity (insulin treatment), whilst lipid lowering control strategies (treatment with statins) seems to be a protective factor. The described methodology has been recently used in the same cohort of patients to describe medical management in coronary T2DM patients with previous percutaneous coronary intervention in Spain, where the EHRead^®^ ability to correctly identify patients’ records containing key variables associated with the study disease is shown [10].

Prior studies have described that around 7.5–15% of the patients with T2DM also had a diagnosis of CAD [23,24], and only 1.78% had a dual diagnosis and no history of stroke, myocardial infarction, or severe liver disease [7]. These patients showed a higher prevalence of comorbidities, with the consequent elevated burden of the disease [7,23,24]. In our study, and in line with previous publications, 1.87% of the patients were identified with both pathologies after applying the described exclusion criteria. Furthermore, we observed arterial hypertension as the main cardiovascular comorbidity present in the most of included patients, followed by heart valve disease, and peripheral vascular disease. Moreover, hyperlipidemia and obesity were also frequently detected. These data correlate with the ones described previously in ATHENA study [25] and using the Ricerca e Salute (ReS) database [7], in which arterial hypertension was detected as the most common comorbidity observed, followed by dyslipidemia.

Due to the chronicity of the diseases and the different comorbidities associated, the management of the patients with diabetes and CAD is challenging, and usually requires a multidisciplinary approach. In 2012, the ACCF/AHA/ACP/AATS/PCNA/SCAI/STS published their guideline for the diagnosis and management of stable CAD with a few specific recommendations for those patients with diabetes [26]. Since then, several advancements in the diagnosis and management of CAD in diabetic patients have occurred. In order to update management evidence, the American Heart Association released in 2020 a scientific statement on the clinical management of stable CAD in patients with T2DM, emphasizing not only recommendations for diabetes or CAD, but also for hyperlipidemia, hypertension and lifestyle management in this group of patients [27]. We found that metformin, the recommended first-line glucose-lowering treatment [28], was the most frequently antidiabetic prescribed drug in our studied population. Moreover, despite clopidogrel, either alone or in combination with ASA, being preferable as antiplatelet therapy to prevent the generalized prothrombotic state options [29], our results showed that ASA alone were most frequently prescribed. On the other hand, we identified that statins were frequently prescribed in these patients in agreement with actual recommendations, which specifies that when diabetes and CAD occur simultaneously, statin therapy is strongly recommended in the primary and secondary prevention of CAD [27]. Additionally, we found that antiplatelet therapy, insulin and b-blockers were frequently prescribed in clinical practice in this population. Finally, although we did not explore therapeutic strategies to improve lifestyle management, the presence of obesity and tobacco use suggest that there is still a need to intervene in this field.

Despite disease management, patients with T2DM and CAD are at high risk of experiencing MACE [30]. In order to address reliable tools to estimate the absolute cardiovascular risk in these patients, several studies have attempted to identify different factors involved [24,31]. Our results showed that coronary disease and T2DM severity, along with other cardiovascular diseases, were independently associated with the development of MACE. Coronary revascularization has been previously described as a protective factor for the development of MACE in patents after the first acute coronary syndrome [32]. Our results in the subpopulation of T2DM with CAD could suggest that DM is not just a comorbidity that affects the development and progression of CAD, but it also has an important role increasing cardiovascular risk per se, even when revascularization occurs.

In line with previous publications, we found that insulin treatment was associated with MACE occurrence [10,33]. Although randomized trials have shown that insulin reduces adverse cardiovascular events [27], in these patients insulin tends to be prescribed in more advanced phases of the disease, which could explain our findings. Finally, other cardiovascular diseases such as heart failure or transient ischemic attack have also been previously associated with the development of MACE [32], although this has not been evaluated in patients with T2DM and CAD. On the other hand, our results confirmed that the use of statins was a protective factor for MACE in the study population. This is in agreement with the large body of evidence, which indicates that statin-based strategies are beneficial lipid-lowering drugs associated with a relative cardiovascular risk reduction in patients with T2DM and CAD [27]. These results, including risk and protective factors, could have an important clinical impact since they could help develop and implement preventive measures for MACE when CAD occurs in diabetic patients, especially those that are modifiable.

## 5. Limitations

There are several limitations to this study. First, our results are based on data captured directly from the unstructured, free-text narratives in patients’ EHRs. Thus, the findings are limited by the availability and accuracy of EHRs, and by the information reported by physicians in their routine practice. Second, because this is a retrospective study based on real-world data, the findings are only hypothesis generating. Moreover, it was not designed to collect variables in a strict, a priori fashion. Thus, there were some interesting variables that were not properly documented and therefore not analyzed. In this regard, the presence of cardiomyopathy at baseline was not analyzed and patients with an implantable cardioverter defibrillator (ICD) were not considered, both possibly related with MACE development [34,35,36,37]. Yet, the effect of some risk factors such as tobacco use may be underreported within the EHRs. Finally, although known confounders were accounted for, unknown confounders might have influenced the findings of the study.

## 6. Conclusions

In conclusion, this study showed that CAD appears in mainly male T2DM patients with other cardiovascular related comorbidities. When it happens, these patients need to be closely monitored because there exists an increasing risk of MACE occurrence. While independent risk factors associated with MACE are difficult to modify since they are intrinsic to the underlying diseases (coronary disease, insulin treatment, transient ischemic attack and heart failure), here we identified statins used as a protective factor that could be used by physicians in clinical practice. Further research is warranted to confirm these findings in future studies.

## Figures and Tables

**Figure 1 jcm-11-06004-f001:**
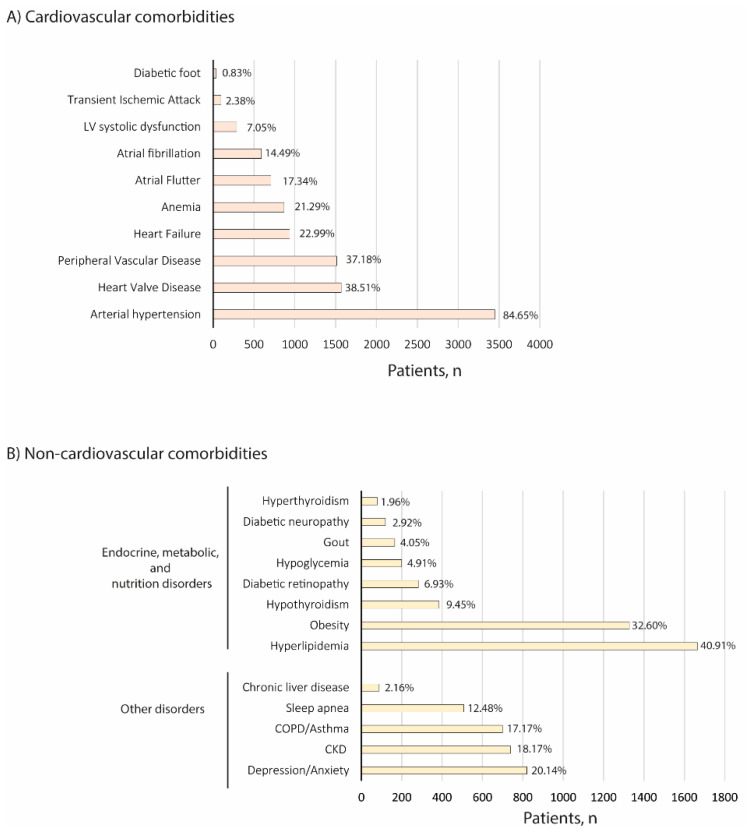
Comorbidities of the study population at index. (**A**) Cardiovascular and (**B**) non-cardiovascular comorbidities are shown. CKD: Chronic kidney disease; COPD: Chronic obstructive pulmonary disease.

**Figure 2 jcm-11-06004-f002:**
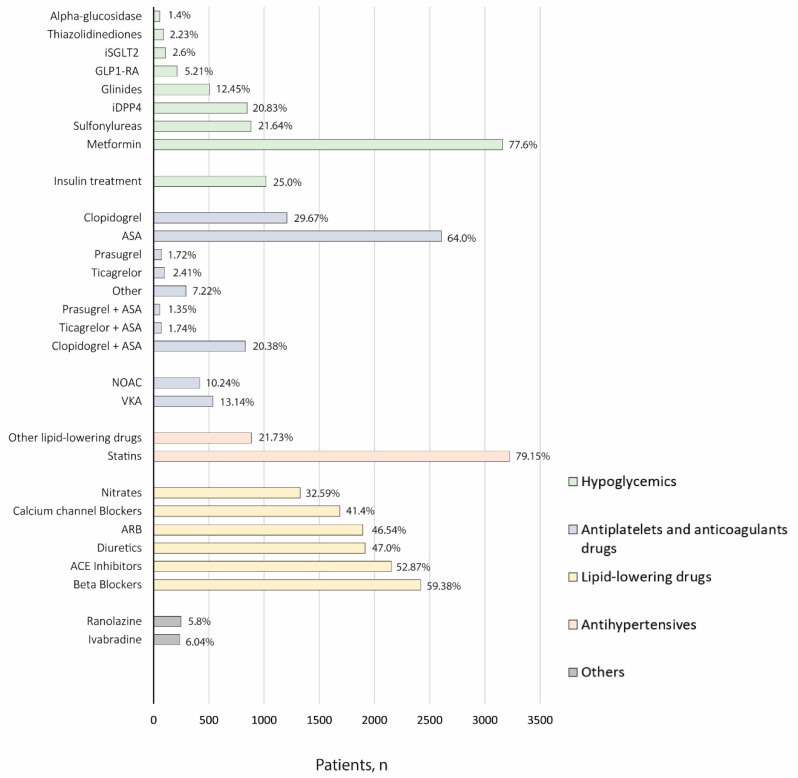
Pharmacological treatment of the study population at index. ACE: Angiotensin-converting enzyme; ARB: Angiotensin II receptor blockers; GLP1-RA: Glucagon-like peptide-1 receptor agonists; iDPP4: Dipeptidil peptidasa-4 inhibitors; iSGLT2: Sodium-glucose cotransporter-2; NOAC: Non-vitamin K antagonist oral anticoagulants; VKA: Vitamin K antagonists.

**Figure 3 jcm-11-06004-f003:**
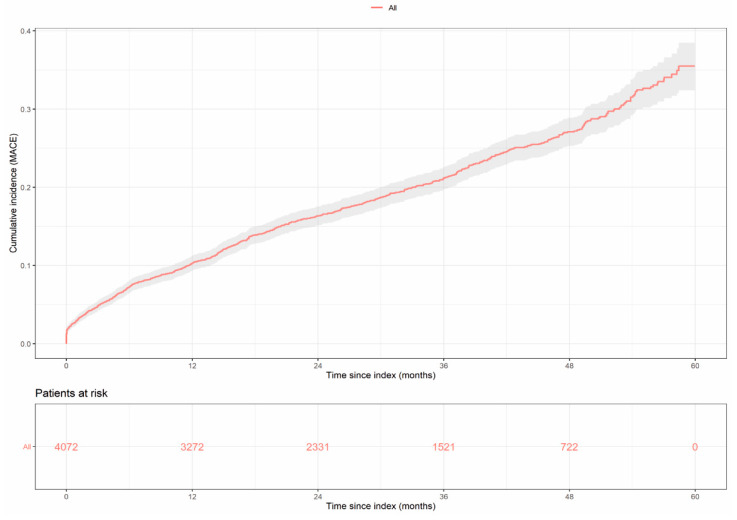
Cumulative incidence of MACE over time during the follow-up period. Probability of any new MACE (myocardial infarction, stroke, unstable angina, and urgent revascularization) is shown. The number of patients at risk across the follow-up period is indicated below.

**Figure 4 jcm-11-06004-f004:**
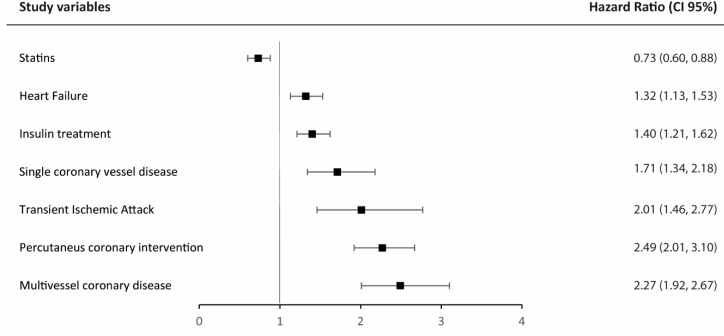
Forest plot of the 7 independent variables associated with the occurrence of MACE during the follow-up in the multivariable model.

**Table 1 jcm-11-06004-t001:** Sociodemographic of the study population at index.

	Study Population (N = 4072)
Age, years	
Mean (SD)	70 (11)
Median (Q1,Q3)	71 (63,78)
Gender, n (%)	
Female	1533 (37.6)
Male	2531 (62.2)
Tobacco use	2208 (54.2)
Current smoker	713 (17.5)
Former smoker	1495 (36.7)
SD: Standard deviation.	

**Table 2 jcm-11-06004-t002:** Cumulative incidence of MACE.

	1-Year FU	2-Year FU	3-Year FU	4-Year FU
MACE *	10.28	16.32	21.18	27.09
Myocardial infarction	4.05	7.8	10.79	13.89
Ischemic stroke	1.77	3.41	5.05	6.94
Unstable angina	1.67	2.49	3.45	4.54
Urgent revascularization	4.04	5.57	6.82	8.69
FU: Follow-up; MACE: Major adverse cardiovascular events.				

* Cumulative incidence of new cases of MACE was estimated through the Kaplan-Meier approach. Each value represents the probability of MACE at different FU time points.

## Data Availability

Not applicable.

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
