# Peer review of "Major Adverse Cardiovascular Events in Coronary Type 2 Diabetic Patients: Identification of Associated Factors Using Electronic Health Records and Natural Language Processing"

_jcm, 2022, doi:10.3390/jcm11206004_

Round 1
Reviewer 1 Report
The extracted Unstructured data from the Electronic Health Records by EHRead. They investigated the association between new MACE and the variables of interest (risk factors, medications, co morbidities) by univariable and multivariable analyses.
Even EHRead is interesting technology based on Natural Language Processing and machine learning, but there are some limitations which are mentioned by the authors at the end of discussion section. Thus, the reliability of results and conclusions in a retrospective study is questionable. In addition, most of the data mentioned in the study is well known in previous studies (as mentioned in the discussion). The study lack novelty. The clinical significance of the study is limited. It does not add important data to the area CAD/DM
Author Response
The extracted Unstructured data from the Electronic Health Records by EHRead. They investigated the association between new MACE and the variables of interest (risk factors, medications, co morbidities) by univariable and multivariable analyses.
Even EHRead is interesting technology based on Natural Language Processing and machine learning, but there are some limitations which are mentioned by the authors at the end of discussion section. Thus, the reliability of results and conclusions in a retrospective study is questionable. In addition, most of the data mentioned in the study is well known in previous studies (as mentioned in the discussion). The study lack novelty. The clinical significance of the study is limited. It does not add important data to the area CAD/DM.
Author’s response: We thank the reviewer for the time invested in reviewing the manuscript. Our results are based on real world data using EHRead technology and natural language processing (NLP) including artificial intelligence and machine learning techniques, a very innovative approach which allowed us the inclusion of 12 hospitals from 6 major regions in Spain. Although this novel technology presents some limitations, all of them described in the manuscript, its usefulness has been previously proven, even in another publication using the same cohort of patients (DOI: 10.1371/journal.pone.0263277). Moreover, limitations are detailed in a specific section in the new version and some changes suggested by other reviewers have been done trying to be as transparent as possible (some new references have been added in the new version). Authors think that these limitations don’t invalidate the results which in turn would be very important when developing and implementing preventive measures for MACE when CAD appear in T2DM patients in routine clinical practice. The importance of our findings has been reinforced in the new manuscript version. Moreover, the insightful comments of reviewers have indeed helped clarified and improved the manuscript.
Reviewer 2 Report
I have several problems with this study. First of all, I believe that there is a strong inclusion bias. Out of over 200 000 patients with diagnosed T2DM only around 4000 were diagnosed with CAD? This is a very low number. Second, the construction of MACE favours the risk factors, which have been found and which therefore are quite intuitive. There is no mortality analysis or hospitalizations for heart failure included. The findings are overall expected and were highly studied before - statins help, more advanced T2DM treated with insulin is a risk factor such as other factors related to MACE components - MVD, PCI, TIA etx. There is no new data in my opinion.
Author Response
I have several problems with this study. First of all, I believe that there is a strong inclusion bias. Out of over 200 000 patients with diagnosed T2DM only around 4000 were diagnosed with CAD? This is a very low number.
Author’s response: We agree with the Reviewer that the number of patients with CAD among those with T2DM may seem low. This may be explained because of the inclusion and exclusion criteria applied to define the study population (detailed in the Materials and methods section). We found 217 632 patients with confirmed diagnosis of T2DM during the study period with at least 6 months of follow-up. Among them, only patients with stable CAD (defined by at least evidence of stenosis > 50% of at least 1 coronary artery but without previous myocardial infarction or stroke) and without planned coronary, cerebrovascular, or peripheral arterial revascularization were included. Moreover, patients with history of liver cirrhosis or liver cancer, intracranial bleeding, renal failure requiring dialysis, or ongoing treatment with anticoagulant medication at index date were excluded. Finally, among patients with T2DM, only those who met all the inclusion criteria and none of the exclusion criteria for CAD were 4072 patients. Finally, as it is explained in Discussion section, these data are in accordance with previous literature which also identified patients with both pathologies (T2DM and CAD) after applying the described exclusion criteria.
Second, the construction of MACE favours the risk factors, which have been found and which therefore are quite intuitive.
Author’s response: We agree that some of the risk factors proposed in this study are related to the definition of MACE. However, it is important to point out that the factors and the outcome (MACE) are not measured in the same time window. All factors are assessed at index (the time of inclusion), but the outcome is analyzed during the follow-up period (there is no overlapping). I.e., we are evaluating new MACE, excluding MACE prior to index. Hence, while some of the factors may be intuitive to clinicians, here we provide evidence of their prognostic value. We believe aligning and complementing the intuition of doctors with the findings derived from scientific research is important and necessary. Furthermore, the use of statins as a protective risk factor in these patients could have a clinical impact not previously described in this scenario.
There is no mortality analysis or hospitalizations for heart failure included.
Author’s response: The Reviewer is right. The objective of this study was to provide real world insights into the clinical characterization and treatment management of diabetic patients with CAD to identify potential risk or protective factors associated with the further occurrence of MACE. Because of that, outcomes as mortality or need of hospitalizations due to heart failure after CAD in T2DM patients would be out of scope and hence, were not included. Future studies should evaluate them as well as the impact of possible preventive measures based on our results.
The findings are overall expected and were highly studied before - statins help, more advanced T2DM treated with insulin is a risk factor such as other factors related to MACE components - MVD, PCI, TIA etx. There is no new data in my opinion.
Author’s response: Our results are based on real world data using EHRead technology and natural language processing (NLP) including artificial intelligence and machine learning techniques which is a very innovative approach allowing the inclusion of 12 hospitals from 6 major regions in Spain. They would be very important when developing and implementing preventive measures for MACE when CAD appear in these patients. In the new version, the importance of these findings has been reinforced. Moreover, the insightful comments of reviewers have indeed helped to clarify and improve the manuscript.
Reviewer 3 Report
Abstract:
Very well-written abstract
Introduction:
-Information in the first paragraph of the introduction is not relevant. I suggest removing it.
-Consider condensing the introduction 2 paragraphs.
Methods:
-Please provide STROBES guidelines for working with retrospective observational studies.
-Consider adding the STROBES guidelines in the supplementary of the paper.
-Has the machine algorithm being used in the study been validated by any prior studies. Please provide references if available.
-Can the authors report what percentage of the missing data was missing? Was it appropriate to remove those cases? Was any imputation done for missing data?
Results: Adequately written and nice figures
Discussion:
Pease expand the limitations section by stating that due to the retrospective nature of the study, the findings are only hypothesis generating
Though known confounders were accounted, for unknown cofounder might have influenced the findings of the study. This must be mentioned in the limitation.
What are the clinical implications of these findings? Please elaborate.
Author Response
Abstract:
Very well-written abstract
Author’s response: We thank the Reviewer for this comment.
Introduction:
-Information in the first paragraph of the introduction is not relevant. I suggest removing it. Consider condensing the introduction 2 paragraphs.
Author’s response: The suggested change has been done.
Methods:
-Please provide STROBES guidelines for working with retrospective observational studies. Consider adding the STROBES guidelines in the supplementary of the paper.
Author’s response: We have included evidence of compliance with the STROBES guidelines as requested (checklist may be found in the Supp. Material).
-Has the machine algorithm being used in the study been validated by any prior studies. Please provide references if available.
Author’s response: The methodology used in this study has been previously used and validated. References 9, 12, 13, 24 and 25 refer to that. Moreover, Reference 10 points to the first published article of the ACORDE study. Here we show the results of the external validation performed for this specific study. We have updated the section “External Validation of EHRead® Performance” and “Discussion” with this reference.
-Can the authors report what percentage of the missing data was missing? Was it appropriate to remove those cases? Was any imputation done for missing data?
Author’s response: We had no missing values for categorical variables, as lack of information (i.e., unavailable data in patients’ EHRs) was considered a ‘true zero’ (e.g., absence of a comorbidity). Regarding numeric variables, we reported the frequency of available data (N in Table S1) without applying any imputation in the descriptive analysis.
However, missing data are not compatible with statistical models such as the Cox PH algorithm used to investigate factors related to MACE. Excepting age (Table 1) -which had no missing information- all numeric variables had > 30% of missing values. As the Reviewer indicates, it wouldn’t be appropriate to remove the patients with fragmentary information (N would significantly drop and we would lose statistical power). Imputing this much missing data could significantly bias the results, especially since missing information may not occur at random (e.g., physicians may report more consistently abnormal laboratory test results than within-range values). Thus, we have resorted to remove the variables with fragmentary information from the inference analysis. We have adapted the “Statistical Data Analyses” section to clarify this.
Results: Adequately written and nice figures
Author’s response: We thank the Reviewer for this comment.
Discussion:
Pease expand the limitations section by stating that due to the retrospective nature of the study, the findings are only hypothesis generating
Author’s response: Suggested changes have been done.
Though known confounders were accounted, for unknown cofounder might have influenced the findings of the study. This must be mentioned in the limitation.
Author’s response: The suggestion has been implemented.
What are the clinical implications of these findings? Please elaborate.
Author’s response: Suggested changes have been done and the importance of these findings when developing and implementing preventive MACE measures in diabetic patients after CAD have been reinforced in the Discussion and Conclusions.
Reviewer 4 Report
I would congratulate the authors for the very good paper. In particular, authors documented 6 risk factors associated with the development of major adverse cardiovascular events (MACE) which were related with cardiovascular diseases and Type 2 diabetes mellitus severity (multivessel and single coronary vessel disease, transient ischemic attack, heart failure, insulin treatment , and percutaneous coronary intervention). On the other hand, treatment with statins was identified as a protective factor for new MACE. The present study uses natural language processing and machine learning to characterize patients with T2DM and CAD in a Spanish real world setting. Results of study are interesting and related to out clinical practice. Here you find my few comments (with some important limitations that should be clarified) in order to improve the manuscript
1.Authors did not consider the presence of cardiomyopathy at baseline, and in limitations I would describe the importance of a previous diagnosis. In particular, hypertrophic cardiomyopathy (HCM) potentially entails thickening of the myocardium and an increased risk of ischaemia. Not by chance coronary artery disease is documented in around 20% of people with HCM, and HCM patients with concomitant CAD are at a higher risk of cardiac death compared with those without coexistent CAD (please cite two references as DOI: 10.1016/j.amjcard.2020.02.002 and 10.1016/j.ijcard.2022.03.028 )
2. In limitations I would also add that ICD patients were not considered (including a population with true risk of sudden cardiac death in both primary and secondary prevention) Not by chance, ICD implantation high-risk population may potentially impact in the long term follow up on multiple aspects (please cite 2 fundamental references: DOI: 10.1007/s40520-018-1088-5 and DOI: 10.1586/14779072.2015.1059276
3. In this scenario, please add a new section entitled "limitations" with all previous suggestions
4. reference list should be updated with few suggested references
Author Response
I would congratulate the authors for the very good paper. In particular, authors documented 6 risk factors associated with the development of major adverse cardiovascular events (MACE) which were related with cardiovascular diseases and Type 2 diabetes mellitus severity (multivessel and single coronary vessel disease, transient ischemic attack, heart failure, insulin treatment , and percutaneous coronary intervention). On the other hand, treatment with statins was identified as a protective factor for new MACE. The present study uses natural language processing and machine learning to characterize patients with T2DM and CAD in a Spanish real world setting. Results of study are interesting and related to out clinical practice. Here you find my few comments (with some important limitations that should be clarified) in order to improve the manuscript
Author’s response: We thank the reviewer for this comment.
1.Authors did not consider the presence of cardiomyopathy at baseline, and in limitations I would describe the importance of a previous diagnosis. In particular, hypertrophic cardiomyopathy (HCM) potentially entails thickening of the myocardium and an increased risk of ischaemia. Not by chance coronary artery disease is documented in around 20% of people with HCM, and HCM patients with concomitant CAD are at a higher risk of cardiac death compared with those without coexistent CAD (please cite two references as DOI: 10.1016/j.amjcard.2020.02.002 and 10.1016/j.ijcard.2022.03.028 )
Author’s response: Although we included several cardiovascular comorbidities at baseline including left ventricle systolic dysfunction, heart failure and arterial hypertension which could be related with cardiomyopathy, the Reviewer is right that we did not specifically include the presence of hypertrophic cardiomyopathy. We have added this aspect in the Limitations section accordingly.
2.In limitations I would also add that ICD patients were not considered (including a population with true risk of sudden cardiac death in both primary and secondary prevention) Not by chance, ICD implantation high-risk population may potentially impact in the long term follow up on multiple aspects (please cite 2 fundamental references: DOI: 10.1007/s40520-018-1088-5 and DOI: 10.1586/14779072.2015.1059276
Author’s response: The suggested changes have been applied.
3.In this scenario, please add a new section entitled "limitations" with all previous suggestions
Author’s response: We thank the Reviewer for this suggestion. Now the manuscript contains a new section entitled “Limitations”.
4.Reference list should be updated with few suggested references
Author’s response: We have added the aforementioned studies as references.
Round 2
Reviewer 1 Report
The author addressed the comments and the test was improved
Reviewer 2 Report
The authors did not modify the manuscript in reply to my comments, so my verdict has not changed.
Reviewer 4 Report
Manuscript definitely improved after reviewer suggestions. Congratulation to authors for the very good work